# ABO Blood Groups in Systemic Sclerosis: Distribution and Association with This Disease’s Characteristics

**DOI:** 10.3390/jcm12010148

**Published:** 2022-12-24

**Authors:** Aurore Collet, Christophe Zawadzki, Emmanuelle Jeanpierre, Caroline Kitel, Sylvain Dubucquoi, Eric Hachulla, Sophie Susen, David Launay

**Affiliations:** 1Institut d’Immunologie, Pôle de Biologie Pathologie Génétique Médicale, CHU Lille, F-59000 Lille, France; 2U1286-Institute for Translational Research in Inflammation (Infinite), Université de Lille, Inserm, CHU Lille, F-59000 Lille, France; 3Laboratoire d’Hémostase, Pôle de Biologie Pathologie Génétique Médicale, CHU Lille, F-59000 Lille, France; 4Service de Médecine Interne et d’Immunologie Clinique, Centre de Référence des Maladies Auto-Immunes et Systémiques Rares du Nord et Nord-Ouest de France (CeRAINO), CHU Lille, F-59000 Lille, France

**Keywords:** systemic sclerosis, blood group, thrombosis, Willebrand factor, factor VIII

## Abstract

Systemic sclerosis (SSc) is an autoimmune disease associated with endothelial activation and fibrosis. Non-O blood group patients carry an increased risk of thrombosis, fibrosis and autoimmune diseases. The aim of our work was to evaluate the distribution of ABO groups in SSc patients and their association with the disease’s characteristics. ABO groups were determined in 504 SSc patients (with 131 completed by a genotypic analysis). The distribution of ABO groups and their diplotypes in SSc patients was comparable to that of the general population, except for haplotypes O1 and B (65.6% vs. 61.6% and 8.8% vs. 5.8% in SSc patients vs. the general population, respectively, *p* = 0.01). The frequency of interstitial lung disease, pulmonary hypertension, calcinosis, digital ulcers, digestive diseases and venous thrombosis, and the Medsger score, were higher in non-O than in O-SSc patients, although they did not display statistical significance. Patients in the non-O group had higher levels of inflammation and endothelial activation biomarkers. In conclusion, the ABO blood group distribution of SSc patients did not differ significantly from that of the general population, but non-O blood groups were associated with inflammation and endothelial activation, and with a non-significant higher frequency of pulmonary and vascular complications in SSc.

## 1. Introduction

The ABO(H) blood group antigens are expressed on red blood cells, but also on many other cells in organs such as the digestive tract, vascular endothelium and platelets, and they could play a role in the pathophysiology of several diseases [1]. Indeed, non-O blood groups are associated with a higher thromboembolic risk, particularly with venous thrombosis [2]. This association may be due to the influence of ABO blood groups on Willebrand factor (VWF) and factor VIII (FVIII) levels, which are higher in non-O blood group individuals [3]. The study of the ABO genotype, determining the number of non-O alleles, allows a more accurate view of the thromboembolic risk than a solely phenotypical ABO study [4]. It was also shown that the A1 allele was associated with a higher risk of thrombosis than the A2 allele, and that the A1 and B alleles were associated with increased FVIII and VWF plasma levels [5]. Non-O blood groups are also associated with fibrosing diseases (liver fibrosis in hepatitis C virus infection [6], keloid scars [7], development of atrio-ventricular blocks linked to the development of intra-cardiac fibrosis [8]). Moreover, the distribution of the ABO blood group in patients with several autoimmune diseases was different from that of the general population [9,10], without a clear pathophysiological explanation.

Among autoimmune diseases, systemic sclerosis (SSc) is a condition characterized by the development of progressive fibrosis of the skin and internal organs, and by vasculopathy, in the context of dysimmunity [11,12]. There is vascular involvement in both the micro- [13,14] and macro-circulation [15]. The pathophysiology of SSc combines (i) vascular damage with the activation of endothelial cells, (ii) the activation of innate and adaptive immune cells with the production of autoantibodies and (iii) the excessive synthesis of the extracellular matrix by fibroblasts, leading to fibrosis [16]. Blood groups may impact these three major pathophysiological components. 

To the best of our knowledge, the repartition of phenotypic blood groups in SSc has been only described in a cohort of 150 SSc patients, showing no difference from that of the Swedish general population [17]. However, there are no data on the association between blood groups and the clinical characteristics of SSc, or on genotypic blood group repartition in SSc, in the literature. We aimed to fill this gap by assessing phenotypic and genotypic blood groups in SSc, and their associations with the characteristics of the disease.

## 2. Materials and Methods

### 2.1. Patient Selection

Prevalent and incident cases of SSc followed in the internal medicine department of the Lille University Hospital were included between January 2019 and January 2022. Eligibility criteria were an age over 18 years and fulfilment of the 2013 ACR/EULAR SSc classification criteria [18].

### 2.2. Collected Data and Methods

Patient demographic data and clinical characteristics of SSc were collected in the Lille University Hospital scleroderma patient database. This database is systematically maintained on an annual basis for all patients followed for SSc at the Lille University Hospital. Disease onset was defined as the time of onset of the first non-Raynaud phenomenon symptom. Interstitial lung disease (ILD) was diagnosed on High-Resolution Computed Tomography of the lung. Pulmonary function tests including forced vital capacity (FVC) and diffusing capacity of the lungs for carbon monoxide (DLCO) were collected. Six-minute walking distance (6MWD) was collected. Pulmonary hypertension (PH) was suspected on a Doppler echocardiogram when the maximum tricuspid regurgitant velocity was >2.8 m/s. Pre-capillary PH was confirmed by right heart catheterization when the mean PAP was found to be ≥25 mm Hg at rest, with mean pulmonary arterial wedge pressure ≤15 mm Hg, using the definition of PH corresponding to the inclusion period. Gastrointestinal tract involvement included esophageal (reflux, dysphagia, stenosis), gastric (gastric vascular ectasia) and intestinal damage (transit disorders, microbial overgrowth of the small intestine, intestinal pseudo-obstruction) and abnormal manometry and/or endoscopy test. 

The patients’ ABO blood groups were first determined via the Etablissement français du sang (EFS) software. If the data were not available in this software, patients were contacted by phone to ask for the blood group registered on their blood group card. Finally, for patients who were newly diagnosed with SSc during the study (incident cases), phenotypical blood group typing was performed during hospitalization for the initial work-up of the disease. 

The ABO genotype of patients was studied by searching for 6 different single-nucleotide polymorphisms (SNPs) in the ABO gene (Table 1), via the allelic discrimination method (Taqman^®^, Thermo Fisher Scientific, Waltham, MA, USA). The ABO patient diplotype (A1O1, O1O1, A1B, etc.) was then defined by combining the 2 alleles carried by the patient.

Endothelial activation and hemostasis parameters were also measured: Factor VIII:C (one-stage assay, Trinoclot aPTT HS, Stago, Asnières sur Seine, France), VWF antigen (VWF:Ag) (immunoturbidimetry, LIAPHEN vWF:Ag, Hyphen Biomed, Neuville-sur-Oise, France), VWF activity (VWF:Act) (platelet binding activity, GPIbM, Innovance^®^ vWF:Act, Siemens, Saint-Denis, France), D-dimers (immunoturbidimetry, Liatest D-DI PLUS, Stago, Asnières sur Seine, France) and fibrinogen (Liquid Fib, Stago, Asnières sur Seine, France).

### 2.3. Ethics

Our study complied with institutional ethical standards and those of the national research committee, the database and biobanking fulfilled the ethical requirements (RCB 2019-A01083-54), and patients provided informed consent. All collected data were anonymized in compliance with French regulations. Data collection and archiving were realized in accordance with the Commission Nationale de l’Informatique et des Libertés (CNIL) guidelines. For genetic analyses, a consent form for participation in RIPH research was signed by the patients.

### 2.4. Statistical Methods

Qualitative variables were described in terms of frequencies and percentages. Quantitative variables were described by their mean and standard deviation, or by their median and interquartile range in the case of a non-Gaussian distribution. The Chi-square test was used to compare the distribution of blood groups to that of the general population [1,19,20]. Patient and disease characteristics were compared between patients of O and non-O groups using the Chi-square test (or Fisher’s exact test if theoretical size <5) for categorical variables and Student’s *t*-test (or Mann–Whitney U test if non-Gaussian distribution) for quantitative variables. The association of VWF and FVIII levels with patient and disease characteristics was assessed using Student’s *t*-test or analysis of variance (ANOVA) for categorical variables, and Pearson’s or Spearman’s correlation coefficient in the case of non-Gaussian distribution for quantitative variables. The significance level was set at 5%. Statistical analyses were performed using SAS software (SAS Institute, version 9.4).

## 3. Results

### 3.1. Patients’ Characteristics

Five hundred and seventy-two patients were identified as satisfying the eligibility criteria. After the exclusion of 68 patients for whom we were unable to retrieve the blood group, or who refused to participate in the study, 504 patients were included in the study. Material was available for the genotyping study for 131 patients (Figure 1).

Demographic, clinical and biological characteristics of the population are detailed in Table 2. Median age at diagnosis was 64 (IQR: 53–73), with a predominance of the female sex (82.1%). Three hundred and thirty-six (66.7%) patients presented limited cutaneous SSc (lcSSc), while 110 (21.8%) presented diffuse cutaneous SSc (dcSSc) and 58 (11.5%) SSc sine scleroderma. Seventy-three (14.5%) patients had a history of venous thrombosis (VT) and 57 (11.3%) had a history of arterial thrombosis (AT). Interstitial lung disease (ILD) was present in 41.8% of cases, and pulmonary arterial hypertension (PAH) in 12.3%. The median modified Rodnan skin score (mRSS) was 3.0 (IQR: 1.0–8.0). Anti-centromere antibodies (ACA) were the most represented auto-antibody (53.4%). Endothelial activation and hemostasis parameters were high compared to the norm: FVIII:C: 191.8 ± 54.5% (N: 50–150%), VWF:Ag: 182.4 ± 73.6% (N: 50–150%) and VWF:Act: 165.3 ± 69.7% (N: 50–150%).

### 3.2. ABO Blood Group Repartition in the SSc Population and Comparison with the General Population

#### 3.2.1. Phenotypical Study

The percentage of SSc patients with the O blood group was 41.6%, those with the A blood group amounted to 41.3%, those with the B blood group amounted to 11.7%, and those in the AB blood group amounted to 5.4% (thus, in total, 58.4% with non-O blood groups). The detailed numbers of patients of each blood type are given in Appendix A. There was no significant difference between SSc patients and the French general population [1] in terms of phenotypical ABO blood group distribution (*p* = 0.06) (Figure 2A) or in terms of O versus non-O distribution (*p* = 0.18) (Figure 2B).

#### 3.2.2. Genotyping Study

We then compared the ABO genotype repartition between SSc patients and the general population. The detailed numbers of patients of each genotype or haplotype are shown in Appendix A. When analyzing the ABO diplotype (combining the paternally and maternally inherited alleles for each patient, n = 131), without distinction between A1/A2 and O1/O2 alleles (Figure 2C, Appendix A), we did not find any difference between SSc patients and the general population in terms of repartition [19] (*p* = 0.36). We then analyzed a more precise diplotype (Figure 2D, Appendix A), with distinction between A1/A2 and O1/O2 alleles (n = 131), and did not observe a difference between SSc patients and the general population in terms of repartition [19] (*p* = 0.31). Finally, we analyzed genotype repartition in terms of haplotype (separating the paternally and maternally inherited alleles for each patient, n = 262) (Figure 2E, Appendix A). Repartition was different between SSc patients and the general population [20]. O1 and B group frequencies were higher in the SSc population than in the general population (65.6% vs. 61.6% and 8.8% vs. 5.8%, respectively, *p* = 0.01), and A1 and A2 group frequencies were lower in the SSc population than in the general population (19.8% vs. 22.5% and 5.0% vs. 6.1%, respectively, *p* = 0.01). 

### 3.3. Association between Phenotypical ABO Blood Group and SSc Characteristics

We then assessed whether ABO blood groups could be associated with the characteristics of the disease (Table 3). As the pro-fibrosing risk associated with blood groups may be associated with their influence on the higher risk of microthrombosis in the non-O blood group [6], we choose to compare SSc characteristics between non-O and O blood group SSc patients. Concerning the clinical characteristics of SSc, blood groups were not associated with any SSc subtype or with the duration of the disease. The presence of ILD, PAH, digital ulcer history, calcinosis and digestive tract involvement was more frequent in the non-O blood type group, but the difference was not statistically significant. Regarding thrombotic risk, VT were more frequent in the non-O blood group (48 (16.3%) vs. 25 (11.9%) in the O blood group), but this was not statistically significant (*p* = 0.16). The percentage of individuals with AT history was comparable in the two groups of patients (10.9% vs. 11.9%, *p* = 0.72). For hemostasis and endothelial activation parameters, the non-O blood group was associated with higher FVIII:C levels (206.5 ± 53.0% vs. 168.1 ± 45.6%, *p* < 0,001), higher VWF:Ag (196.2 ± 73.5% vs. 160.4 ± 68.6%, *p* < 0.001) and higher VWF:Act (181.4 ± 74.3% vs. 141.1 ± 54.1%, *p* < 0.001). Concerning inflammation, non-O blood groups were associated with a higher CRP (4.2 ± 8.8 mg/L vs. 3.5 ± 8.2 mg/L, *p* = 0.021). Concerning ANA specificities, blood groups were not associated with ACA or with anti Scl70 auto-antibodies (*p* = 0.52 and 0.51, respectively). The percentage of anti-RNAP3-positive patients was lower in the non-O blood group (1.6 vs. 4.9%, *p* = 0.04). Blood groups were not associated with the severity of cardiopulmonary involvement in SSc. Indeed, ILD extension at CT-scan, FVC, DLCO, NYHA dyspnea scale, 6 MWT, triscupid leak and NT-pro-BNP were comparable between non-O and O blood group patients. Only FEV/FVC was lower in the non-O blood group than in the O blood group (76.9 ± 7.8 vs. 78.5 ± 8.6, *p* = 0.039). Finally, the severity scores of the disease (EUSTAR 2011 and EUSTAR 2016) did not differ between non-O and O blood group patients, but the Medsger score almost reached statistical significance (0.8 ± 1.8 vs. 0.5 ± 1.5 in non-O vs. O patients, respectively, *p* = 0.051).

### 3.4. Association between Haemostasis and Endothelial Activation Parameters and SSc Characteristics

Finally, as we found that some SSc complications (ILD, PAH, digestive tract involvement) were more frequent in patients in the non-O blood group, we sought to determine whether this association could be driven by the influence of the blood group on hemostasis dysregulation and endothelial activation. We therefore studied the association between VWF and FVIII levels and some characteristics of the disease (Table 4). Concerning the clinical parameters of the disease, in patients with ILD, endothelial activation parameters were higher than in patients without ILD (VWF:Ag: 193.4 ± 78.2 vs. 176.1 ± 69.3, *p* = 0.08; VWF:Act: 177.6 ± 79.2 vs. 156.4 ± 59.5, *p* = 0.05; FVIII:C: 199.0 ± 54.8 vs. 187.7 ± 53.1, *p* = 0.12). VWF:Ag and VWF:Act were positively correlated with mRSS (*p* = 0.02 and *p* = 0.003, respectively).

Concerning thrombotic risk, VWF:Ag and FVIII:C were significantly higher in patients with than in patients without a history of AT (244.0 ± 120.2% vs. 176.5 ± 64.9%, *p* = 0.02; 239.2 ± 81.7% vs. 187.4 ± 49.4%, *p* = 0.01, respectively). Concerning inflammation, CRP was positively correlated with VWF:Ag (*p* = 0.0004), VWF:Act (*p* = 0.01) and FVIII:C (*p* = 0.0004). Regarding the severity of cardiopulmonary involvement in SSc, VWF:Ag was positively correlated with tricuspid leak (*p* = 0.04) and NT-pro-BNP (*p* = 0.0008). VWF:Ag and VWF:Act were negatively correlated with DLCO (*p* = 0.005 and *p* = 0.01, respectively). Finally, the EUSTAR 2011 score was positively correlated with VWF:Act (*p* = 0.03) and the EUSTAR 2016 score was positively correlated with VWF:Ag (*p* = 0.01) and with VWF:Act (*p* = 0.003).

## 4. Discussion

The first objective of our study was to describe the phenotypic and genotypic distribution of ABO blood groups in a cohort of French SSc patients, and to compare this to the general population. We did not observe any difference in the distribution of blood groups between our cohort of SSc patients and the general population, with the exception of the haplotype distribution, with a higher frequency of O1 and B alleles in SSc patients. These results are in line with the Swedish study performed by Fabianne and Nordin [17]. In this study, the authors found that phenotypical ABO blood group repartition in 150 SSc patients was close to that in the Swedish general population. Thus, contrary to what was suggested for various other autoimmune [9,10,21,22] and fibrosing diseases [6,7,8], we did not find, at a phenotypic level, an under-representation of the O blood group in SSc.

Our study was the first to investigate the association of the ABO blood group with the clinical and biological characteristics of SSc. Interstitial lung disease, PAH, calcinosis and digestive involvement were consistently more frequent, and the Medsger score was slightly higher (0.8 vs. 0.5, *p* = 0.051), in patients in the non-O group, although the differences were not statistically significant. This may suggest that ABO groups may be partly associated with organ involvement in this disease. In the literature, it has been shown that non-O blood groups are more frequently found in patients with post-embolic pulmonary hypertension [23,24]. Overall, our results are comparable to those found in a recent study on lupus, in which ABO blood group repartition was similar between lupus patients and the general population, but with a higher rate of complications (autoimmune hemolytic anemia, arthritis) in B blood group patients [25].

We also found a significantly higher CRP level in non-O SSc patients than in O patients. This is in line with previous studies showing that higher CRP levels in non-O patients were also found in other conditions, such as ischemic heart disease [26]. A SNP (rs8176704) in the ABO gene has been shown to be associated with higher CRP levels [27]. GWAS studies have also found associations between different ABO gene variants and the levels of certain pro-inflammatory cytokines, such as IL6 and TNFα [27,28]. A higher CRP and a higher frequency of ILD in non-O SSc patients are in line with the association between CRP and ILD, as shown in a previous work by our team [29]. 

The presence of an increased risk of VT in the non-O group has been recognized for many years in the literature [30], with an odds ratio of 2.44 compared to the O group in a meta-analysis [31]. In our cohort of SSc patients, the presence of a history of VT was more frequent in the non-O group, although this was not significant. Some studies also suggest an increased risk of AT associated with the non-O blood group [31,32,33]. However, in our study, the percentage of patients with a history of AT was comparable between group O and non-O patients. The excess thrombotic risk described in the literature in non-O patients might be explained by the higher levels of FVIII and VWF in this population [34]. Indeed, in non-O individuals, VWF is less degraded due to the longer oligosaccharide chains on its surface, which protect it from proteolysis [2]. In our cohort of SSc patients, levels of VWF:Ag, VWF:Act and FVIII:C were indeed higher in non-O patients than in O patients. This could explain the higher VT rate in our non-O patients. Recently, it was also shown that the A blood group was associated with higher levels of soluble forms of the soluble VEGF receptor, which compete with the membrane-bound VEGF receptor for the binding of VEGF, thereby inhibiting the outgrowth of blood vessels. These results are consistent with the higher risk of acute cardiovascular events observed in A blood type individuals relative to those of type O [35].

Finally, to better understand the ways in which blood groups may influence SSc complications, we assessed VWF and FVIII levels in SSc and their associations with disease characteristics. We found that these markers of endothelial activation were elevated relative to the norms, as reported by numerous studies [36,37,38]. Moreover, the levels of these endothelial activation markers seemed to be associated with organ involvement, such as ILD and PAH, and with greater severity (DLCO, tricuspid leak, NT-pro-BNP). These associations are known in the literature [39,40], and they reinforce our hypothesis that the higher frequency of SSc complications in non-O patients may be driven by their influence on hemostasis and endothelial activation. In one study, serum VWF levels predicted future pulmonary arterial pressure elevation when using a logistic regression method [41]. In our work, VWF levels were also positively correlated with mRSS, an association that, to our knowledge, has not been described in the literature. Endothelial activation parameters were positively correlated with EUSTAR 2011 and 2016 severity scores, which is consistent with an association with the severity of skin involvement, and with pulmonary and cardiac complications. This association of VWF levels with the severity of SSc has also been described [42].The association of elevated VWF and FVIII with excessive thrombotic risk is well known in the general population [2,31] and has also been described in SSc for VT [43]. To our knowledge, our study is the first to report elevated VWF and FVIII levels in SSc patients with a history of AT. A recent study showed that the high FVIII levels found in SSc were predictive of an enhanced thrombin generation potential, which might contribute to the higher risk of thromboembolic events in this disease [44]. However, in our study, given their association with CRP, we cannot exclude the possibility that endothelial activation levels’ correlations with the clinical parameters of SSc and a history of thrombosis are explained by inflammation, rather than by direct causality.

Our study has limitations. First, our study was purely exploratory and descriptive. The results that we found should be confirmed by mechanistic and functional data. Next, we obtained no data on the different patterns of capillaroscopic alterations in our cohort, precluding any study on their associations with ABO blood groups. Furthermore, we cannot exclude the possibility that our results, derived from the French population, are not extendable to other populations. Indeed, it is known that the distribution of blood groups differs according to ethnicity [45,46], a criterion that we were unable to take into account in our work. It is also important to note that for the diplotype study, we compared our cohort of SSc patients to a German blood donor population [19,20], which was the closest population to the French population for which we could find a description of the ABO diplotype distribution in the literature. It is possible that this population is not entirely similar to the French general population. Moreover, for the genotyping study, our number of patients for whom material was available was smaller (n = 131) and therefore we cannot exclude the possibility that there was a lack of statistical power. Due to this low number of genotyping data, we could not study the associations among the ABO genotype and SSc characteristics. Finally, most of the associations that we found between endothelial activation levels and SSc parameters were weak, although statistically significant. In addition, as our study was exploratory, we used a large number of statistical tests, which may have led to multiple comparison issues.

## 5. Conclusions

Our work found that, in a large cohort of SSc patients, there was no major difference in the distribution of ABO blood groups compared to the general population, either in phenotypic or diplotype genotypic terms. Unlike in other fibrosing and autoimmune diseases, blood groups do not appear to play a major role in SSc. On the other hand, non-O blood groups appeared to be associated with inflammation and endothelial activation, and with a non-significantly higher frequency of pulmonary and vascular complications in SSc. While having no direct clinical impact, this study addresses an important question on the distribution and associations of ABO blood groups in SSc.

## Figures and Tables

**Figure 1 jcm-12-00148-f001:**
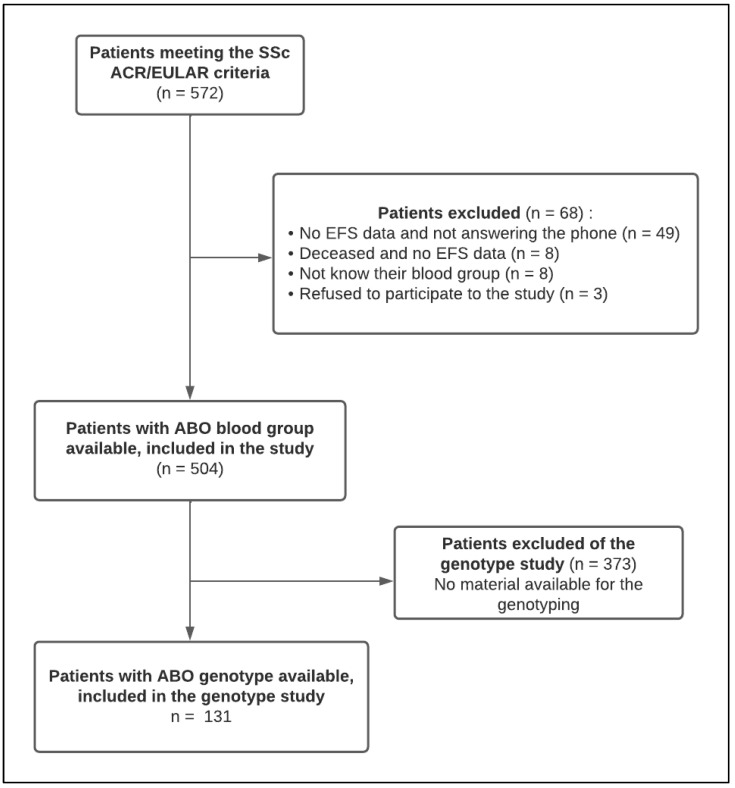
Participant flow chart. EFS: Etablissement français du sang.

**Figure 2 jcm-12-00148-f002:**
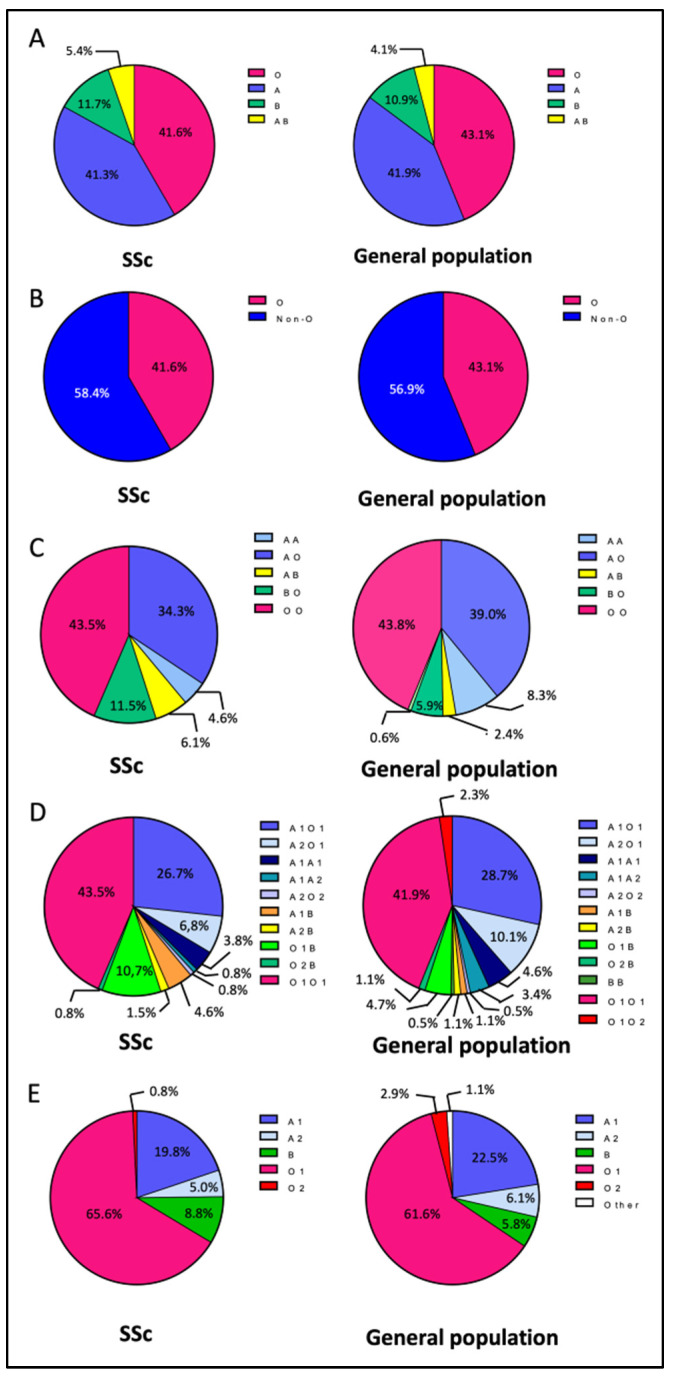
ABO blood group repartition in the SSc cohort and in the general population. (**A**) Phenotypical ABO blood group repartition in the SSc population compared to that of the French general population [1]. (**B**) O vs. non-O phenotypical blood group repartition in the SSc population compared to that of the French general population [1]. (**C**) Simplified diplotype blood group repartition in the SSc population compared to that of the general population [19]. (**D**) Diplotype blood group repartition in the SSc population compared to that of the general population [19]. (**E**) Haplotype blood group repartition in the SSc population compared to that of the general population [20]. SSc: systemic sclerosis.

**Table 1 jcm-12-00148-t001:** List of the SNPs used to define the ABO genotype.

Allele	SNP
O1	rs8176719
A1/O2	rs579459
B	rs8176749
A1	rs2519093
A1/A2	rs1053878
O2	rs41302905

SNP: single-nucleotide polymorphism.

**Table 2 jcm-12-00148-t002:** Patients’ characteristics.

Demographic Parameters	
Age—median (Q1; Q3), year	64 (53; 73)
Women—n (%)	414 (82.1)
**SSc Clinical Characteristics**	
SSc subtype	
dcSSc—n (%)	110 (21.8)
lcSSc—n (%)	336 (66.7)
SSc sine scleroderma—n (%)	58 (11.5)
Time since SSc diagnosis—median (Q1; Q3), year	10 (6; 19)
VT history—n (%)	73 (14.5)
AT history—n (%)	57 (11.3)
ILD—n (%)	204 (41.8)
ILD extension	
Limited—n (% among ILD)	140 (68.6)
Extensive—n (% among ILD)	52 (25.5)
ND—n (% among ILD)	12 (5.9)
PAH—n (%)	62 (12.3)
Digital ulcer, past or present—n (%)	245 (49.1)
Calcinosis—n (%)	88 (17.8)
Digestive tract involvement—n (%)	60 (11.9)
Renal crisis—n (%)	7 (1.4)
mRSS—median (Q1; Q3)	3.0 (1.0; 8.0)
Telangiectasias—n (%)	378 (75.8)
NYHA score	
I—n (%)	232 (46.4)
II—n (%)	135 (27.0)
III—n (%)	92 (18.4)
IV—n (%)	41 (8.2)
**Biological Parameters**	
CRP—median (Q1; Q3), mg/L	0.0 (0.0; 5.0)
NT-pro-BNP—median (Q1; Q3), pg/mL	107.0 (52.0; 252.0)
ANA specificity	
ACA—n (%)	234 (53.4)
Scl70—n (%)	85 (19.4)
RNAP3—n (%)	13 (3.0)
RNP—n (%)	9 (2.1)
**Hemostasis and Endothelial Activation Parameters**	
FVIII:C—mean ± sd (%)	191.8 ± 54.5
VWF:Ag—mean ± sd (%)	182.4 ± 73.6
VWF:Act—mean ± sd (%)	165.3 ± 69.7
D dimers—median (Q1; Q3), ng/mL	360.0 (270.0; 590.0)
Fibrinogen—mean ± sd (g/L)	3.6 ± 0.9
**Cardiopulmonary Involvement Evaluation**	
Tricuspid regurgitant velocity—mean ± sd (m/s)	2.6 ± 0.6
FVC—mean ± sd (% of predicted value)	100.3 ± 22.1
FEV/FVC—mean ± sd	77.5 ± 8.1
DLCO—mean ± sd (% of predicted value)	67.6 ± 19.7
Distance at 6 MWT—mean ± sd (% of predicted value)	75.5 ± 19.0
**Activity and Severity of the Disease**	
EUSTAR 2011 score—median (Q1; Q3)	1.0 (0.5; 2.0)
EUSTAR 2016 score—median (Q1; Q3)	1.2 (0.2; 1.9)
Medsger score—median (Q1; Q3)	0.0 (0.0; 1.0)
Death—n (%)	7 (1.4)

ACA: anti-centromere antibody; AT: arterial thrombosis; CRP: C-reactive protein; DLCO: diffusion capacity of the lung for carbon monoxide; EUSTAR: European Scleroderma Trials and Research Group; FVIII:C: factor VIII; FVC: forced vital capacity; FEV: forced expiratory volume; ILD: interstitial lung disease; mRSS: modified Rodnan skin score; NYHA: New York Heart Association; PAH: pulmonary arterial hypertension; VT: venous thrombosis; VWF:Act: Willebrand factor activity; VWF:Ag: Willebrand factor antigen; 6 MWT: 6 min walking test.

**Table 3 jcm-12-00148-t003:** Associations between ABO blood groups and SSc characteristics.

	Non-O Blood Group (n = 294)	O Blood Group (n = 210)	*p*
Demographic Parameters			
Age—median (IQR), year	64 (53–73)	64 (54–74)	0.30
Women—n (%)	238 (81)	176 (83)	0.41
SSc Clinical Characteristics			
SSc subtype			0.77
dcSSc—n (%)	61 (20.7)	49 (23.3)	
lcSSc—n (%)	198 (67.3)	138 (65.7)	
SSc sine scleroderma—n (%)	35 (11.9)	23 (11.0)	
Delay since SSc diagnosis—median (Q1; Q3), year	10.0 (6.0; 18.0)	11.0 (6.0; 19.0)	0.41
VT history—n (%)	48 (16.3)	25 (11.9)	0.16
AT history—n (%)	32 (10.9)	25 (11.9)	0.72
ILD—n (%)	122 (42.8)	82 (40.4)	0.59
ILD extension			0.11
Limited—n (% among ILD)	90 (31.1)	50 (24.0)	
Extensive—n (% among ILD)	25 (8.7)	27 (13.0)	
PAH—n (%)	41 (15.3)	21 (11.2)	0.21
Digital ulcer—n (%)	151 (51.7)	84 (45.4)	0.17
Calcinosis—n (%)	55 (19.2)	33 (15.9)	0.36
Organic microangiopathy—n (%)	88 (62.8)	17 (70.8)	0.49
Digestive tract involvement—n (%)	38 (12.9)	22 (10.5)	0.40
Renal crisis—n (%)	4 (1.4)	3 (1.5)	NA
mRSS—median (Q1; Q3)	4.0 (2.0; 8.0)	3.0 (0.0; 7.0)	0.44
Telangiectasias—n (%)	224 (77.2)	154 (73.7)	0.36
NYHA score			0.20
I—n (%)	128 (44.1)	104 (49.5)	
II—n (%)	88 (30.3)	47 (22.4)	
III—n (%)	49 (16.9)	43 (20.5)	
IV—n (%)	25 (8.6)	16 (7.6)	
Biological Parameters			
CRP—median (Q1; Q3), mg/L	0.0 (0.0; 5.0)	0.0 (0.0; 4.0)	**0.021**
NT-pro-BNP—median (Q1; Q3), pg/mL	103.0 (51.0; 248.0)	115.0 (58.0; 259.0)	0.52
ANA specificity			
ACA—n (%)	139 (54.7)	95 (51.6)	0.52
Scl70—n (%)	52 (20.5)	33 (17.9)	0.51
RNAP3—n (%)	4 (1.6)	9 (4.9)	**0.04**
RNP—n (%)	3 (1.2)	6 (3.3)	0.18
Hemostasis and Endothelial Activation Parameters			
FVIII:C—mean ± sd (%)	206.5 ± 53.0	168.1 ± 48.6	**<0.001**
VWF:Ag—mean ± sd (%)	196.2 ± 73.5	160.4 ± 68.6	**<0.001**
VWF:Act—mean ± sd (%)	181.4 ± 74.3	141.1 ± 54.1	**<0.001**
D dimers—median (Q1; Q3), ng/mL	350 (270.0; 540.0)	370.0 (270.0; 620.0)	0.28
Fibrinogen—mean ± sd (g/L)	3.6 ± 0.8	3.6 ± 1.0	0.93
Cardiopulmonary Involvement Evaluation			
Tricuspid leak—mean ± sd (m/s)	2.7 ± 0.6	2.6 ± 0.5	0.098
FVC—mean ± sd (% of predicted value)	100.7 ± 22.3	99.8 ± 21.7	0.65
FEV/FVC—mean ± sd	76.9 ± 7.8	78.5 ± 8.6	**0.039**
DLCO—mean ± sd (% of predicted value)	68.1 ± 20.0	67.0 ± 19.3	0.54
Distance at 6 MWT—mean ± sd (% of predicted value)	75.8 ± 19.2	75.1 ± 18.9	0.72
Severity of the Disease			
EUSTAR 2011 score—median (Q1; Q3)	1.0 (0.5; 2.0)	1.0 (0.5; 2.0)	0.43
EUSTAR 2016 score—median (Q1; Q3)	1.2 (0.3; 1.9)	1.2 (0.2; 2.0)	0.73
Medsger score			0.051
0—n (%)	186 (70.7)	142 (78.5)	
1—n (%)	27 (10.3)	17 (9.4)	
2—n (%)	17 (6.5)	10 (5.5)	
3—n (%)	16 (6.1)	3 (1.7)	
4—n (%)	5 (1.5)	4 (2.2)	
5—n (%)	2 (0.8)	1 (0.6)	
6—n (%)	3 (1.1)	2 (1.1)	
7—n (%)	4 (1.5)	1 (0.6)	
8—n (%)	1 (0.4)	0 (0.0)	
9—n (%)	1 (0.4)	0 (0.0)	
10—n (%)	2 (0.8)	0 (0.0)	
13—n (%)	0 (0.0)	1 (0.6)	

ACA: anti-centromere antibody; AT: arterial thrombosis; CRP: C-reactive protein; DLCO: diffusion capacity of the lung for carbon monoxide; EUSTAR: European Scleroderma Trials and Research Group; FVIII:C: factor VIII; FEV: forced expiratory volume; FVC: forced vital capacity; ILD: interstitial lung disease; mRSS: modified Rodnan skin score; NYHA: New York Heart Association; PAH: pulmonary arterial hypertension; VT: venous thrombosis; VWF:Act: Willebrand factor activity; VWF:Ag: Willebrand factor antigen; 6 MWT: 6 min walking test.

**Table 4 jcm-12-00148-t004:** Associations between hemostasis and endothelial activation parameters and SSc characteristics.

	VWF:Ag % n = 219	VWF:Act % n = 171	FVIII:C % n = 226
Age (year) Pearson correlation coefficient *p*	0.18 **0.006**	0.17 **0.02**	0.30 **<0.0001**
Sex (Women vs. Men) *p*	179.7 ± 75.8 vs. 193.4 ± 63.0 0.27	159.7 ± 69.8 vs. 186.0 ± 65.8 **0.04**	192.4 ± 56.6 vs. 189.4 ± 45.7 0.73
SSc subtype (lcSSc vs. dcSSc vs. sine scleroderma) *p*	175.9 ± 66.5 vs. 195.6 ± 72.3 vs. 189.5 ± 101.9 0.22	156.3 ± 67.67 vs. 186.7 ± 63.6 vs. 165.8 ± 83.3 0.05	185.6 ± 52.19 vs. 198.3 ± 46.4 vs. 210.27 ± 73.03 0.05
VT history (Presence vs. Absence) *p*	197.0 ± 96.0 vs. 180.3 ± 69.9 0.38	190.2 ± 79.3 vs. 161.5 ± 67.6 0.07	201.3 ± 59.1 vs. 190.5 ± 53.9 0.33
AT history (Presence vs. Absence) *p*	244.0 ± 120.2 vs. 176.5 ± 64.9 **0.02**	213.3 ± 143.2 vs. 161.3 ± 58.9 0.21	239.2 ± 81.7 vs. 187.4 ± 49.4 **0.01**
ILD (Presence vs. Absence) *p*	193.4 ± 78.2 vs. 176.1 ± 69.3 0.08	177.6 ± 79.2 vs. 156.4 ± 59.5 0.05	199.0 ± 54.8 vs. 187.7 ± 53.1 0.12
PAH (Presence vs. Absence) *p*	195.0 ± 61.8 vs. 182.8 ± 76.0 0.48	188.6 ± 63.8 vs. 163.2 ± 71.6 0.17	195.2 ± 45.7 vs. 192.9 ± 55.8 0.86
Digital ulcer (Presence vs. Absence) *p*	183.0 ± 84.1 vs. 182.0 ± 62.6 0.92	165.8 ± 80.0 vs. 165.0 ± 57.8 0.94	187.2 ± 54.0 vs. 196.9 ± 54.9 0.18
mRSS Pearson correlation coefficient *p*	0.16 **0.02**	0.23 **0.003**	0.10 0.13
Telangiectasias (Presence vs. Absence) *p*	182.7 ± 73.6 vs. 183.0 ± 75.1 0.98	168.6 ± 72.4 vs. 156.3 ± 60.8 0.34	193.1 ± 55.2 vs. 189.1 ± 53.6 0.64
ACA specificity (Presence vs. Absence) *p*	177.0 ± 59.7 vs. 189.0 ± 82.3 0.25	159.8 ± 70.4 vs. 172.3 ± 68.7 0.28	194.5 ± 54.3 vs. 190.7 ± 53.6 0.62
Anti Scl70 specificity (Presence vs. Absence) *p*	187.1 ± 71.7 vs. 182.0 ± 72.2 0.69	171.6 ± 65.1 vs. 164.4 ± 71.0 0.61	200.8 ± 54.7 vs. 190.4 ± 53.6 0.28
Tricuspid leak (m/s) Pearson correlation coefficient *p*	0.16 **0.04**	0.17 0.05	0.15 0.05
FVC (% of predicted value) Pearson correlation coefficient *p*	−0.16 **0.02**	−0.12 0.12	−0.08 0.22
DLCO (% of predicted value) Pearson correlation coefficient *p*	−0.19 **0.005**	−0.19 **0.01**	−0.12 0.08
CRP (mg/L) Pearson correlation coefficient *p*	0.24 **0.0004**	0.19 **0.01**	0.24 **0.0004**
Nt-pro-BNP (pg/mL) Pearson correlation coefficient *p*	0.23 **0.0008**	0.19 **0.01**	0.12 0.08
Medsger score Pearson correlation coefficient *p*	0.06 0.40	0.02 0.80	0.09 0.22
EUSTAR 2011 score Pearson correlation coefficient *p*	0.09 0.19	0.18 **0.03**	−0.04 0.52
EUSTAR 2016 score Pearson correlation coefficient *p*	0.20 **0.01**	0.26 **0.003**	0.11 0.14

For qualitative variables, results are expressed as mean ± SD in each group, and *p* value. For quantitative variables, results are expressed as Pearson coefficient and p value. ACA: anti-centromere antibody; AT: arterial thrombosis; CRP: C-reactive protein; DLCO: diffusion capacity of the lung for carbon monoxide; EUSTAR: European Scleroderma Trials and Research Group; FVC: forced vital capacity; FVIII:C: factor VIII; ILD: interstitial lung disease; mRSS: modified Rodnan skin score; PAH: pulmonary arterial hypertension; VT: venous thrombosis; VWF:Act: Willebrand factor activity; VWF:Ag: Willebrand factor antigen.

## Data Availability

Not applicable.

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
