# Peer review of "ABO Blood Groups in Systemic Sclerosis: Distribution and Association with This Disease’s Characteristics"

_jcm, 2022, doi:10.3390/jcm12010148_

Round 1

Reviewer 1 Report

This study aims to evaluate the distribution of ABO groups in SSc patients and their association with clinical manifestations of the disease. The topic covered is very interesting and new. Some aspects of this article may open new perspectives in SSC pathogenesis research and clinical management.  

My only comment is this: one of the major SSc clinical manifestations is the secondary Raynaud's phenomenon, accompanied by capillaroscopic alterations. A more severe pattern seems to be associated with more severe organ manifestations. Did the authors consider the association of the parameters evaluated with the various capillaroscopic patterns? It's not mentioned in the text and tables. 

Reviewer 2 Report

In this manuscript, Collet and coworkers report the distribution and clinical associations of ABO blood groups in a cohort of patients with systemic sclerosis. The results of the study show that ABO blood group distribution is similar in SSc patients and the general population, but non-O groups appear to have a more inflammatory phenotype and endothelial activation.

Overall, the study fills a gap in the knowledge with regard to blood groups and SSc. However, there are some limitations that should be underlined:

- My main concerns are: 1. most of the findings (e.g. the association of non-O groups with hemostasis and endothelial activation, higher endothelial activation in SSc with inflammatory/progressive features) are widely known; 2. most of the association parameters (e.g. correlations coefficients) are weak despite statistical significance and this should be underlined; 3. the association of endothelial activation markers with SSc clinical parameters may rather be explained by inflammation (see CRP) and does not imply causality on thrombosis or complications of SSc; 4. there could be a multiple comparisons issue given the large number of statistical tests.

In my opinion the study would benefit from identifying a single main objective and reporting only the distribution of blood groups and its associations.

- The manuscript, and particularly the Results section, is sometimes difficult to read. Additionally, english needs spell and grammar checks (e.g. consistently used instead of significantly).

Reviewer 3 Report

The study by Collet and collaborators has been well conducted and described but, in my opinion, the provided results have low clinical relevance. The major limit of this work is that it is purely descriptive, lacking any mechanistic or functional data strenghtening the results.

The authors say that this is the first study investigating the association of ABO blood groups with the clinical and biological characteristics of SSc patients: however, they have found few statistically significant associations. Moreover, most data are already known in literature.

Discussion: English style must be improved.

Round 2

Reviewer 2 Report

The authors recognized the limitations of the study and improved the manuscript as suggested.